# New Era for Next-Generation Sequencing in Japan

**DOI:** 10.3390/cancers11060742

**Published:** 2019-05-28

**Authors:** Masayuki Takeda, Kazuko Sakai, Takayuki Takahama, Kazuya Fukuoka, Kazuhiko Nakagawa, Kazuto Nishio

**Affiliations:** 1Department of Medical Oncology, Kindai University Faculty of Medicine, 377-2 Ohno-higashi, Osaka-Sayama, Osaka 589-8511, Japan; takahama_t@med.kindai.ac.jp (T.T.); kfukuoka@med.kindai.ac.jp (K.F.); nakagawa@med.kindai.ac.jp (K.N.); 2Department of Genome Biology, Kindai University Faculty of Medicine, 377-2 Ohno-higashi, Osaka-Sayama, Osaka 589-8511, Japan; kasakai@med.kindai.ac.jp (K.S.); knishio@med.kindai.ac.jp (K.N.); 3Clinical Research Center, Kindai University Faculty of Medicine, 377-2 Ohno-higashi, Osaka-Sayama, Osaka 589-8511, Japan

**Keywords:** next-generation sequencing (NGS), clinical sequencing, solid cancer, precision medicine

## Abstract

Recent progress in understanding the molecular basis of cancer—including the discovery of cancer-associated genes such as oncogenes and tumor suppressor genes—has suggested that cancer can become a treatable disease. The identification of driver oncogenes such as *EGFR*, *ALK*, *ROS1*, *BRAF* and *HER2* has already been successfully translated into clinical practice for individuals with solid tumor. Next-generation sequencing (NGS) technologies have led to the ability to test for multiple cancer-related genes at once with a small amount of cells and tissues. In Japan, several hospitals have started NGS-based mutational profiling screening in patients with solid tumor in order to guide patients to relevant clinical trials. The Ministry of Health, Labor, and Welfare of Japan has also approved several cancer gene panels for use in clinical practice. However, there is an urgent need to develop a medical curriculum of clinical variant interpretation and reporting. We review recent progress in the implementation of NGS in Japan.

## 1. Introduction

Insight into the molecular basis of cancer including the discovery of cancer-associated genes such as oncogenes and tumor suppressor genes has set cancer on the path to becoming a treatable disease. The recent identification of driver oncogenes, such as *EGFR* [1,2,3], *ALK* [4,5,6,7], *ROS1* [8], and *BRAF* [9,10] has already been successfully translated into clinical practice for individuals with lung cancer. The introduction of genomic biomarkers to guide targeted therapy has also had marked success for patients with *BRAF* mutation-positive melanoma [11] or *HER2*-overexpressing gastric or breast cancer [12,13]. There is ethnic difference on the frequencies of mutations of *EGFR* and *KRAS* between Asian and Caucasian compared to other solid tumors [14]. Given that companion diagnostic testing, as exemplified by the phrase “one companion diagnostic–one drug”, has been a standard of the Pharmaceuticals and Medical Devices Agency (PMDA) of Japan for the detection of somatic mutations in certain types of cancer, the development of targeted therapies in relation to the mechanisms of action of the corresponding gene alterations has highlighted the need to detect actionable driver mutations in a small tumor sample. Next-generation sequencing (NGS) has emerged as a new technology for the performance of multiple cancer susceptibility genes simultaneously with a small amount of tissue. In the case of lung cancer, NGS has been found to be effective for the detection of actionable mutations in a small amount of tumor tissue, enabling patient selection for genotype-based therapies [15,16,17]. This technology can also significantly reduce sequencing costs and turn-around time for a genetic test. Since 2013, several institutions including the National Cancer Center (NCC) and university hospitals in Japan have initiated research-based NGS clinical sequencing in order to allow the matching of investigational drugs or approved targeted agents to patients with corresponding molecular alterations. Educational programs have been developed to help disseminate knowledge about clinical sequencing among medical professionals, and practice guidance for NGS gene-panel testing of solid tumors has recently been published. Oncologists are thus now able to take advantage of several cancer gene panels that have been approved for clinical practice by The Ministry of Health, Labor, and Welfare (MHLW) of Japan. This review describes the recent NGS-based clinical sequencing projects in Japan and discusses issues relating to the integration of NGS into patient care.

## 2. Genetic Testing Guidelines for Lung, Breast, and Gastric Cancer

Among the solid malignant tumor types, non-small cell lung cancer (NSCLC) has become a prominent example of the application of precision medicine. New clinical practice guidelines for the treatment of lung cancer have been published by the Japan Lung Cancer Society in 2018 (Figure 1), with the first decision step for advanced NSCLC being based on (1) the detection of known oncogenic drivers including *EGFR* or *BRAF* mutations or *ALK* or *ROS1* rearrangements; (2) the expression of programmed cell death-ligand 1 (PD-L1) in the tumor at a threshold level of ≥50%; and (3) the absence of driver mutations and a PD-L1 expression level of <50% or of unknown status. Although *BRAF* testing was not included in the previous Japan Lung Cancer Society guidelines, it was added in 2018, given that dabrafenib in combination with trametinib has now been established as a standard of care option for NSCLC patients with *BRAF* mutation [9,10]. The cost of testing for the abovementioned driver mutations and PD-L1 expression is paid in part by the medical care system in Japan. As of March 2019, companion diagnostic testing for other oncogenic drivers such as *RET* and *NTRK* rearrangements has not been approved. With regard to the targeted treatment of metastatic breast and gastric cancer, clinical practice guidelines are currently based only on *HER2* status, although olaparib is recently approved for patients with metastatic breast cancer harboring germ line *BRCA* mutation.

## 3. Research-Based NGS Panels

### 3.1. SCRUM-Japan

In February 2013, a nationwide cancer genomic screening project (LC-SCRUM-Japan) was initiated for the application of personalized medicine to advanced NSCLC. As of May 2016, more than 200 institutions across Japan had joined this program. LC-SCRUM-Japan was extended to become a multiorgan program, SCRUM-Japan, which currently consists of LC-SCRUM-Japan and GI (gastrointestinal)–SCREEN. Tumor specimens are now analyzed with the quality-assured Oncomine Comprehensive Assay version 3 (OCA v3, Thermo Fisher Scientific, Waltham, MA, USA), which enables simultaneous analysis of DNA and RNA to detect hundreds of variant types, including single nucleotide variants (SNVs), copy number variations (CNVs), gene fusions, and indels from unique cancer driver genes across 161 genes relevant to solid tumors. In addition, SCRUM-Japan has recently extended its testing program to include comprehensive liquid biopsy with the Guardant 360 assay, which evaluates 73 cancer-related genes in cell-free tumor DNA present in blood specimens. Such screening of patients with advanced lung or gastrointestinal cancer allows them to be matched with approved drugs and experimental therapies in clinical trials without the need for an invasive tissue biopsy.

A phase II study (LURET) of vandetanib in patients with *RET*-rearranged advanced NSCLC was performed in conjunction with SCRUM-Japan [18]. Among the 1536 screened NSCLC patients, 9 of 17 (53%) eligible *RET* rearrangement-positive individuals evaluable for the primary analysis showed an objective response, with no severe adverse events. These results suggested that vandetanib is effective and safe for the treatment of individuals with *RET*-rearranged advanced NSCLC. The SCRUM-Japan program is also currently facilitating patient enrolment into a clinical trial of alectinib. Given that this drug, a more selective inhibitor of anaplastic lymphoma kinase (ALK) compared to crizotinib and ceritinib, also shows a high level of inhibitory activity toward RET, the efficacy of alectinib in *RET* fusion-positive NSCLC patients is being evaluated in a phase I/II trial (ALL-RET) [19].

### 3.2. Clinical Sequencing at Our Hospital

In July 2013, NGS testing was initiated at our institution, Kindai University Hospital, to make an appropriate medical decision about access to approved drug or investigational drugs for lung cancer patients. Small gene panels—the Ion AmpliSeq RNA Fusion Lung Cancer Research Panel (Thermo Fisher Scientific) and Ion AmpliSeq Colon and Lung Cancer Panel (Thermo Fisher Scientific), with simultaneous sequencing of *ALK*, *RET*, *ROS1*, and *NTRK1* fusion transcripts and hotspot and targeted regions of 22 genes implicated in colon and lung cancers, respectively—were applied prospectively [15]. Patients harboring an actionable genetic alteration who have successfully accessed a targeted agent were found to have a longer survival compared to those positive for an actionable mutation who did not receive targeted therapy. Limitations also exist with respect to clinical trial design such as retrospective, nonrandomized study.

The increasing availability of NGS data can provide insight into the clinical background of patients with tumors positive for rare driver mutations. We focused on NSCLC patients with exon-20 mutations of *EGFR* or *HER2* identified by clinical sequencing, and evaluated their clinical characteristics, and treatment outcomes. We found that epidermal growth factor receptor (EGFR)-tyrosine kinase inhibitors (TKIs) and nivolumab showed limited efficacy in patients with exon-20 mutations of *EGFR* or *HER2*, respectively, and that other potentially actionable genetic alterations rarely coexist with these mutations [20].

### 3.3. MSK-IMPACT Platform in Academia

Several academic institutions in Japan carried out a high-throughput DNA sequencing project, MSK-IMPACT in 2016. This assay involves hybridization capture and deep sequencing of all protein-coding exons of 468 cancer-associated genes, with the assay being performed in a Clinical Laboratory Improvement Amendments (CLIA) certified laboratory [21]. Tumor and nontumor specimens were sent from the participating hospitals to the MSK Cancer Center in the United States, with the median processing time from obtaining informed consent from the patient to providing the genotyping results to the treating physician being ~1.5 months.

## 4. Approved NGS Panels in Japan

### 4.1. NCC Oncopanel

The NCC in Japan has developed an original sequencing panel, the NCC Oncopanel. This panel was designed with the use of the application SureDesign (Agilent Technologies, Santa Clara, CA, USA) to capture mutations or amplifications of 114 genes including 12 gene fusions with the aim of identifying actionable genetic alterations for selection of suitable molecularly targeted therapies (Table 1). The panel allows the analysis of both tumor and matched normal tissue or blood specimens for both somatic and germ line mutations. Sequencing libraries are prepared with the use of the SureSelect XT reagent (Agilent Technologies). Variants can be filtered on the basis of a read depth of ≥100× and a variant allele frequency threshold of ≥5%. Gene amplification is defined as a log_2_ fold change in read depth of ≥2. Although genes in the NCC Oncopanel include those known to be somatically altered in solid tumors, the panel does not have companion diagnostic indications for any specific target drug.

### 4.2. FoundationOne CDx

FoundationOne CDx is an NGS-based in vitro diagnostic device for detection of genomic alterations of 324 genes known to drive cancer growth, providing potentially actionable information to help guide treatment options (Table 1). It also provides information on genomic signatures including microsatellite instability and tumor mutational burden, the latter of which is based on the total number of all synonymous and nonsynonymous variants present at an allele frequency of ≥5%. FoundationOne CDx is a U.S. Food and Drug Administration (FDA) approved companion diagnostic for the identification of patients with certain molecular subtypes of NSCLC (*EGFR* mutation, *ALK* rearrangement, or *BRAF* mutation), melanoma (*BRAF* mutation), colorectal cancer (*KRAS* or *NRAS* mutation), ovarian cancer (*BRCA1* or *BRCA2* mutation), or breast cancer (*HER2* amplification) who might benefit from targeted therapy. In December 2018, MHLW of Japan approved FoundationOne CDx as a comprehensive genomic profiling test for all solid tumors and a broad companion diagnostic for individuals with advanced cancer. In contrast to the FDA-approved conditions, *BRAF* mutation in NSCLC was excluded from the companion diagnostic designation in Japan.

### 4.3. Oncomine Dx Target Test

In June 2017, the FDA approved Oncomine Dx Target Test (Thermo Fisher Scientific), which simultaneously evaluates 46 driver gene variants with as little as 10 ng of DNA or RNA (Table 1), as the first NGS-based companion diagnostic for screening of NSCLC tumor specimens. In April 2018, the test was approved in Japan as a companion diagnostic for the identification of NSCLC patients who might benefit from therapies that target the V600E mutation of *BRAF*. In December 2018, it was granted reimbursement coverage for the *BRAF* mutation indication as a companion diagnostic for dabrafenib in combination with trametinib. In February 2019, the MHLW of Japan granted expanded approval for the use of the test to detect exon-19 deletions and the L858R point mutation of *EGFR* as well as *ALK* fusions and *ROS1* fusions.

## 5. NGS Guidance and Educational Programs

Although there has been no standard procedure for handling tumor tissue for clinical sequencing, the Japanese Society of Pathology has set the rule for handling of pathological tissue samples for genomic medicine such as concentration of formaldehyde and fixation time to provide quality assurance and quality control

In the United States, to establish standardized classification, interpretation, annotation, and reporting of sequence variants associated with cancer, a joint consensus recommendation regarding “Standards and Guidelines for the Interpretation and Reporting of Sequence Variants in Cancer” was published in January 2017 [22]. No similar guidelines or guidance for cancer genomic testing currently exists in Japan, despite government initiatives to promote the use of NGS for cancer patients in clinical practice. Guidance for gene-panel testing of solid cancers—in particular, childhood cancers, rare cancers, and carcinomas of unknown primary site—was published in October 2017 as the result of a collaboration of the Japanese Society of Medical Oncology (JSMO), Japanese Society of Clinical Oncology (JSCO), and Japanese Cancer Association (JCA) [23]. According to the guidance, NGS panel can be used for precise prediction, diagnosis, and prognostic assessment. It is provided on the premise that, when an approved companion diagnostic is available, such as in the case of NSCLC and colorectal cancer, physicians should consider the established test as priority. The levels of evidence for detection of individual genomic alterations by gene-panel testing in Japan should not differ greatly from the definitions standardized in the United States and European Union [22]. The definitions adopted in this guidance were developed as appropriate for the medical system in Japan, with evidence levels related to therapeutic efficacy being classified as follows: level 1A, PMDA-approved companion diagnostic biomarker for the tumor type; level 1B, FDA-approved companion diagnostic biomarker (or complementary diagnostic biomarker) for the tumor type, an adequately powered, prospective study with biomarker selection or stratification, or a meta-analysis showing that the drug is clinically effective in a biomarker-selected cohort of patients with the same tumor type; level 2A, biomarker that is associated with response of the same tumor type to the drug as shown by subgroup analysis in a prospective clinical trial; level 2B, approved biomarker in the other tumor type or biomarker with evidence of clinical efficacy; level 3A, a single or few unusual responders, or case studies, suggesting a biomarker is associated with response to the drug and with the association being supported by a scientific rationale; level 3B, preclinical data (in vitro or in vivo models, or functional genomics) showing that a biomarker predicts a cellular response; and level 4, other gene mutations in cancer.

The complexity of precision medicine delivery to oncology patients in the context of the national public health insurance scheme has highlighted the need for molecular tumor boards (MTBs), also known as “expert panels” in Japan, to assess when molecular testing for tumor profiling is appropriate and to address the therapeutic opportunity of approved drug or investigational new drugs. MTBs consist of medical professionals such as medical oncologists, a molecular biologist, pathologists, a geneticist, a genetic counselor, and a nurse navigator. However, there has been no standardized educational program regarding curation and annotation systems for MTBs consisting of multidisciplinary cancer specialists. The Japan Agency for Medical Research and Development supports some research programs to promote the education of medical professionals with regard to clinical cancer sequencing. These programs consist of lectures, tutorial sessions, and e-learning (https://www.e-precisionmedicine.com/en/medicine).

## 6. Cancer Genome Database Project

In 2017, the MHLW of Japan convened a Roundtable Consortium on the Promotion of Cancer Genomic Medicine. The consortium set the direction for genomic medicine in Japan with regard to providing citizens with early access to the world’s leading-edge cancer genome-based medicines by capitalizing on the country’s strengths to develop innovative treatment methods for Japanese and other Asians through integration of knowledge of cancer genomic medicine and creation of a framework to lead the field on a global scale. In February 2018, 11 institutions across Japan were selected as core hospitals for cancer genomic medicine by the MHLW. In June 2018, the Center for Cancer Genomics and Advanced Therapeutics (C-CAT) was launched by the NCC to develop a cancer genomics information repository, a master database for cancer genomic medicine. The C-CAT plays four key roles to facilitate the implementation of cancer genomic medicine in Japan: (1) To advance and control the quality of genomic diagnosis of cancer by securing Japanese clinical and genomic information at domestic public institutions, creating a Cancer Knowledge DataBase (CKDB) optimized for Japan, contributing to decision making by expert panels at core hospitals for cancer genomic medicine, communicating information on cancer genomic medicine to the Japanese public, and advocating policies underpinned by data accumulated nationwide. (2) To promote both data sharing among core hospitals according to appropriate protocols as well as cancer care covered by the national health insurance. (3) To promote clinical trials as well as research and development by providing a robust foundation for clinical studies and investigator-initiated clinical trials based on core data, serving as a resource for drug development and personalized medicine, and opening avenues to private corporations. (4) To conduct research and nurture talent in preparation for the introduction of whole-genome sequencing to health care. The collection of cancer genomic data linked to clinical information such as, outcome of conventional chemotherapy or prognosis from a large number of individuals can provide historical control data for patients harboring rare mutations when novel targeted therapies are assessed for prospectively selected individuals with tumors harboring these genomic alterations.

## 7. Ongoing or Planned Basket Trials for Rare Genetic Variants

The U.S. National Cancer Institute’s NCI-MATCH (Molecular Analysis for Therapy Choice) trial is a phase II basket study that was launched in August 2015. It will enroll ~1000 patients with any solid tumor or lymphoma assigned to treatment arms based on the molecular profiles of their disease.

A few basket trails are also ongoing in Japan. The receptor tyrosine kinase RET can be oncogenically activated as a result of gene fusions or point mutations. *RET* fusions occur in a variety of malignancies, including 1% to 2% of lung cancers and 10% to 20% of papillary thyroid cancers [24]. LOXO-292 is an oral and selective investigational drug in clinical development for the treatment of cancers with *RET* abnormalities, with phase II trials of this drug currently enrolling such patients in Japan [25]. Entrectinib is a highly potent, orally available, ATP-competitive TKI that has shown inhibitory efficacy in vitro at subnanomolar concentrations against NTRK1, NTRK2, NTRK3, ROS1, and ALK [26]. Phase I studies of entrectinib in patients with advanced or metastatic solid tumors showed that 19 (79%) of the 24 evaluable patients harboring rearrangements of *NTRK1/2/3*, *ROS1*, or *ALK* who constituted the phase II–eligible population, achieved a response [27]. On the basis of these promising results, STARTRK-2, an open-label, multicenter, global phase II basket study of entrectinib for the treatment of patients with solid tumors positive for *NTRK1/2/3*, *ROS1*, or *ALK* fusions, is now under way.

The Master Key project is a platform trial consisting of registry and substudy parts for rare cancer patients in Japan. The project was launched in April 2017 with collaboration between academia and pharmaceutical companies. In the substudy part, a basket trial is planned for which the eligibility criteria are specified not by tumor type but by the presence of a specific biomarker.

## 8. Issues Surrounding NGS Panels

Increased access to NGS-based genetic testing for patients in Japan has raised several concerns. Under the health care system in Japan, the insured pay 30% and insurers pay 70% of medical costs. Given that expensive cancer drugs such as immune checkpoint inhibitors and agents matched to genomic alterations have recently been approved for a variety of cancers, the government needs to control the overall expense of procedures and medications by limiting their indications. NGS testing is thus limited in principle to once for each patient according to recently published guidance [23]. This situation necessitates selection of the most appropriate NGS panel for each patient among those that have been approved.

Given that the development of a custom NGS panel and sequencer needs technological innovations as well as a substantial investment, most institutions in Japan are dependent on the validated ready-made vendor solutions from outside of Japan, such as those made by Thermo Fisher and Illumina. The country has thus fallen behind in developing a domestic NGS panel and promoting such innovation by Japanese industry. Furthermore, there is a tremendous need to develop novel targeted agents matched with tumor molecular alteration, such as NCI-MATCH in the United States.

## 9. Conclusions

NGS panel can evaluate multiple genes at the same time with a small amount of tissue. Moreover, it can assist physicians in the matching of cancer patients found to harbor actionable genetic alterations with approved or investigational drugs. Since the result of a randomized trial demonstrated that matched therapy provided no advantage in progression-free survival (PFS) compared to physician’s choice therapy [28], further advance will require the real-time acquisition of knowledge of genetic alterations that can support clinical decision-making.

## Figures and Tables

**Figure 1 cancers-11-00742-f001:**
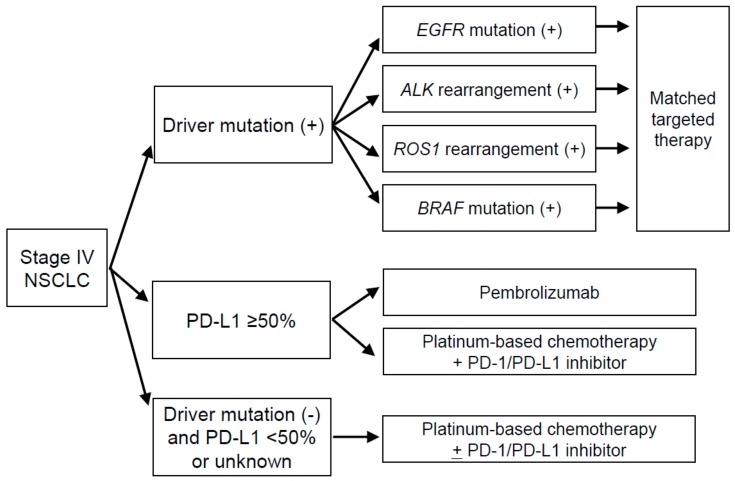
Treatment strategy for stage IV non-small cell lung cancer (NSCLC).

**Table 1 cancers-11-00742-t001:** Approved next-generation sequencing (NGS) panels in Japan.

	Oncomine Dx Target Test	FoundationOne CDx	NCC Oncopanel
No. of genes	46	324	114
Sample	DNA/RNA	DNA	DNA
Paired sample (control)			Blood
FDA approval	Yes	Yes	No
PMDA approval	Yes	Yes	Yes
Companion diagnostic	Yes	Yes	No

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
