# Peer review of "New Era for Next-Generation Sequencing in Japan"

_cancers, 2019, doi:10.3390/cancers11060742_

Reviewer 1 Report

Figure 1 (clinical practice guideline for the treatment of lung cancer) is missing. Please include it.

Author Response

I am terribly sorry that figure is not included. Accordingly, I have now added it.

Reviewer 2 Report

Takeda et al provide a brief overview about the implementation of NGS diagnostic tests in oncology in Japan. I did not have access to the figure 1 in the submitted paper.

The paper is well written, but is not very interesting for a reader outside Japan.

It could have been really more interesting to discuss some technical points, such as :

- the technical limitations of NGS diagnostic tests (quantity of material available, quality of the DNA),

- the problem of spatial and temporal heterogeneity of cancers (intrapatient clonal heterogeneity)

- the medico-economics problems : accessibility to a specialized lab all around the country, delay to obtain the diagnosis,

- bioinformatics pipelines

- the next step : WES, RNAseq, WGS...

Minor points :

line 97-100 : please be careful with the conclusions of 1 retrospective study, when prospective clinical trials (Shiva) failed to demonstrate a clinical benefice of this approach.

Author Response

We thank the reviewer for the positive assessment of our paper. Our responses to the specific points raised are as follows:

Although there has been no standard procedure for handling tumor tissue for clinical sequencing, the Japanese Society of Pathology have set the rule for handling of pathological tissue samples for genomic medicine such as concentration of formaldehyde and fixation time to provide quality assurance and quality control. These informations have now been addressed in the revised manuscript (p. 8, lines 7-10).

As reviewer pointed out, intrapatient heterogenity, medico-economical problem, bioinformatical issue, and WES are very important issue for handling NGS in cancer patients. However, these infrastructure is not arranged in japan yet. Therefore, in this review, we foused the already arranged infrastructure regarding NGS system.

Accodingly, the precion medicine is under depelopment since the result of a randomized trial (SHIVA trial) demonstrated that matched therapy provided no advantage in progression-free survival (PFS) compared with physician's choice therapy. These informations have now been addressed in the revised manuscript (p. 12, lines 10-12).  

Reviewer 3 Report

This review highlights aspects of introducing NGS-based clinical sequencing Projects  into clinical practice in Japan.

Author Response

Thank you for your positive comments

Reviewer 4 Report

The manuscript by Takeda et. al. sheds light on the newly adopted use of NGS panels and personalized medicine in Japan. They enumerate the NGS panels which are approved in Japan such as NCC oncopanel, FoundationOne CDx and Oncomine Dx Target test and discuss their relevance. They also stress the requirement for NSG guidance and educational programs in Japan. Overall, the manuscript serves as a good source for brief commentary on the available options for patients in terms on personalized medicine.

Comments

1.     Although the authors justifiably point out the lack of an indigenous Oncopanel with companion diagnostics, they do not sufficiently point out the requirement for it. Could difference in tumor mutation spectrum between diverse population from USA and EU and Japan, make the international oncopanel less suitable for widespread application as the authors point out the cost associated limits one test per patient.

2.     The subheading 1.1 is misleading as it mainly focuses on Lung Cancer treatment guideline with a cursory note on breast and gastric cancer.

3.     There was no associated figure although the text refers to one in line 51.

It might be very useful if the manuscript provided some statistics of the oncopanel use or genetic testing either nationally or the Kindai University Hospital.

Author Response

We thank the reviewer for the positive assessment of our paper. Our responses to the specific points raised are as follows:

As reviewer pointed out, there is ethnic difference on the frequencies of mutations of EGFR and KRAS between Asian and Caucasian compared to the other solid tumor. These informations have now been addressed in the revised manuscript (p. 3, lines 8-10). In other solid tumor, there is no significant difference on the distribution of actionable mutation between asian and caucasian. Thus, there is no need to develop of original NGS panel although development of custom panel may help to promote innovation by Japanese industry.

Accodingly, we have now provided the information of guidline of breast and gastric (p.4, lines 17-20).

I am terribly sorry that figure is not included. Accordingly, I have now added it.

Round  2

Reviewer 2 Report

Thanks for the clarifications. As I said in my first review, the paper is well written, but the overall interest can be low outside of Japan. As a reviewer, I can not decide if the paper is in the editorial line of Cancers.

Reviewer 3 Report

-